# Breast Cancer Treatment: The Case of Gold(I)-Based Compounds as a Promising Class of Bioactive Molecules

**DOI:** 10.3390/biom12010080

**Published:** 2022-01-05

**Authors:** Rossana Galassi, Lorenzo Luciani, Junbiao Wang, Silvia Vincenzetti, Lishan Cui, Augusto Amici, Stefania Pucciarelli, Cristina Marchini

**Affiliations:** 1Chemistry Division, School of Science and Technology, University of Camerino, 62032 Camerino, Italy; lorenzo.luciani@unicam.it; 2School of Biosciences and Veterinary Medicine, University of Camerino, 62032 Camerino, Italy; junbiao.wang@unicam.it (J.W.); silvia.vincenzetti@unicam.it (S.V.); lishan.cui@unicam.it (L.C.); augusto.amici@unicam.it (A.A.); stefania.pucciarelli@unicam.it (S.P.)

**Keywords:** gold, breast cancer, Auranofin, phosphane compounds, carbene compounds, IC_50_, in vivo, in vitro, metal-based drugs, molecular targets

## Abstract

Breast cancers (BCs) may present dramatic diagnoses, both for ineffective therapies and for the limited outcomes in terms of lifespan. For these types of tumors, the search for new drugs is a primary necessity. It is widely recognized that gold compounds are highly active and extremely potent as anticancer agents against many cancer cell lines. The presence of the metal plays an essential role in the activation of the cytotoxicity of these coordination compounds, whose activity, if restricted to the ligands alone, would be non-existent. On the other hand, gold exhibits a complex biochemistry, substantially variable depending on the chemical environments around the central metal. In this review, the scientific findings of the last 6–7 years on two classes of gold(I) compounds, containing phosphane or carbene ligands, are reviewed. In addition to this class of Au(I) compounds, the recent developments in the application of Auranofin in regards to BCs are reported. Auranofin is a triethylphosphine-thiosugar compound that, being a drug approved by the FDA—therefore extensively studied—is an interesting lead gold compound and a good comparison to understand the activities of structurally related Au(I) compounds.

## 1. Introduction

Breast cancer is one of the most common forms of tumors for women and some subtypes of breast cancers are still without efficacious therapy. The search for new drugs with improved and wider efficacy is a current challenge. The present review focuses on the most relevant results reported in the scientific literature published over the last 6–7 years about the in vitro and/or in vivo treatment of breast cancers by two classes of gold(I)-based drugs, phosphane and N-Heterocyclic Carbene (NHC) compounds, and on the main mechanism of actions thereby disclosed. The results are discussed comparatively to the classic metal-based drug, cisplatin, and to Auranofin.

## 2. Breast Cancer: An Introduction to the Disease

### Breast Cancers, Current and Potential Metal-Based Alternative Therapies

Breast cancer is the most common cancer in women worldwide. Advances in early detection and therapy have resulted in significant improvement in breast cancer survival rates, achieving survival probabilities of 90% for at least 5 years after diagnosis in most developed countries. However, the 5-year survival rate drops to 27% in case of metastatic disease, which is considered barely curable with currently available therapies. For patients without metastatic disease, therapeutic goals are tumor eradication, by surgery and radiation therapy, and prevention of recurrences by systemic adjuvant therapies. Different treatment strategies are selected according to the molecular subtype. Breast cancer is a heterogeneous disease that can be classified, based on hormone and human epidermal growth factor receptor 2 (HER2) status, into three major subtypes: luminal (expressing the estrogen receptor (ER) and/or progesterone receptor (PR)), that can be further divided in luminal A (ER+/PR+/HER2−) and luminal B (ER+/PR+/HER2+); HER2-overexpressing (ER−/PR−/HER2+); and triple-negative (TNBC), lacking expression of ER, PR and HER-2 [1]. Approximately, 75% of all breast cancers express hormone receptors; thus, they can be treated with endocrine therapies, alone or in addition to chemotherapy. Estrogen-induced activation of ERs promotes proliferation and survival of both normal and cancerous breast tissue through transcription of pro-survival genes and activation of cellular signaling. Endocrine therapies include selective ER modulators (e.g., tamoxifen), that compete with estrogen for binding to ER; selective ER downregulators (e.g., fulvestrant), that induce ER protein degradation or block ER transcriptional activity; and aromatase inhibitors, that suppress estrogen production by blocking the conversion of androgens to estrogens [2]. HER2-positive breast cancer accounts for 20–25% of all breast cancers and is associated with a worse prognosis. Multiple HER2-targeted therapies have been approved for the treatment of HER2-positive breast cancer as monotherapies or in combination with chemotherapies and they have significantly improved survival outcomes for this type of breast cancer patients. HER2-targeted therapies include monoclonal antibodies, such as trastuzumab and pertuzumab; antibody-drug conjugates, such as trastuzumab emtansine (T-DM1); and tyrosine kinase inhibitors, such as lapatinib and neratinib. Although both endocrine- and HER2-targeted therapies have considerably reduced breast cancer recurrence and mortality, de novo and acquired resistance to these treatments is a serious concern [3]. TNBC accounts for approximately 15–20% of all breast cancers and, clinically, it represents a particularly aggressive subtype with a high proliferative rate. In contrast to the other breast cancer subtypes that are most common in middle-aged and older women, TNBC is more prevalent in younger women (<50 years). Among risk factors associated with TNBC, there is the presence of a mutation in breast cancer susceptibility genes, BRCA1 and BRCA2 [4]. Because TNBC lacks ER, PR and HER-2 expression, it is not sensitive to endocrine or HER-2-targeted therapy. Although new treatment options are emerging, including phosphoinositide 3-kinase (PI3K) pathway inhibitors, immune checkpoint inhibitors and cyclin-dependent kinase (CDK) inhibitors, standard chemotherapy used to treat metastatic breast cancer remains the main therapeutic option for TNBC [5]. Conventional chemotherapy includes taxanes (docetaxel or paclitaxel), which inhibit microtubule depolymerization leading to mitotic arrest, or drugs that cause DNA damage, such as fluorouracil, anthracyclines (doxorubicin) and alkylating agents (cyclophosphamide). TNBC appears to be more sensitive to chemotherapy, but it is correlated with a higher rate of early recurrence and distant metastasis with respect to other breast cancer subtypes [6]. Patients with BRCA1/2 mutations can receive poly(ADP-ribose) polymerase (PARP) inhibitors or platinum-based (cisplatin or carboplatin) chemotherapy, which exploit the DNA repair defects in tumors with BRCA1/2 mutations [7]. The PARP enzyme plays a critical role in the repair of DNA single-strand breaks through the base-excision repair (BER) mechanism. In the presence of BRCA1 and BRCA2 mutations, which affect homologous recombination repair of double-stranded DNA breaks, PARP inhibition results in cell death, because the cells are unable to repair DNA damage by either BER or homologous recombination.

The major limitations of currently available chemotherapeutics are their toxic side effects, which considerably decrease the quality of life of cancer patients and cancer drug resistance. Thus, there is an urgent need to develop innovative drugs to treat breast cancer with improved clinical efficacy and tolerability. In this scenario, gold-based compounds display a promising anticancer activity and they might represent an alternative chemotherapeutic strategy to other well-known metal-based anticancer drugs such as cisplatin for the treatment of various human cancers, including breast cancer.

The toxicity of cisplatin and its derivatives (oxaliplatin and carboplatin) can be mainly explained by the lack of selectivity for cancer cells concerning normal tissues. Cisplatin can induce cell death through nuclear DNA binding and this primary mode of action is not only limited to cancer cells but also normal cells [8]. In the last years, ruthenium complexes have emerged as potential metal-based drug candidates to replace platinum chemotherapy, since they seem to exert their anticancer effects through multitarget mechanisms and display fewer side effects and favorable clearance properties. The water-soluble ruthenium(II) complex [Ru(p-cymene)(bis(3,5-dimethylpyrazol-1-yl)methane)Cl]Cl (UNICAM-1), for instance, showed significant anticancer activity in a preclinical model of TNBC and resulted to be considerably better tolerated than cisplatin [9]. UNICAM-1 exhibited a clear mitochondrial target, exerting its anticancer effects by promoting impairment of mitochondrial functionality [10]. Moreover, UNICAM-1 was able to reverse tumor-associated immune suppression by significantly reducing the number of tumor-infiltrating regulatory T cells [9]. Among ruthenium complexes, NAMI-A ((ImH)[trans-RuCl4(dmso-S)(Im)], Im = imidazole) and KP1019/1339 (KP1019 = (IndH)[trans-RuCl4(Ind)2], Ind = indazole; KP1339 = Na[trans-RuCl4(Ind)2]), two structurally related ruthenium(III) coordination compounds, have attracted attention for their promising pharmacological performances and for having reached clinical trials. KP1019 is known mainly as a cytotoxic agent for the treatment of platinum-resistant colorectal cancers, whereas NAMI-A has successfully entered clinical trials as an antimetastatic drug [11]. However, the results of clinical investigations were not encouraging, probably because the limited stability in physiological conditions reduced their therapeutic effectiveness [12]. The major drawback of these compounds is the fact that their mechanism of action is not fully understood. Hence, there is still a need to develop novel metal-based anticancer agents, alternatives to platinum-based drugs, with known modes of action and molecular targets.

## 3. Gold(I)-Based Drugs in the Treatment of Breast Cancer

### 3.1. Gold(I)-Based Compounds

#### 3.1.1. Auranofin

Auranofin is an FDA-approved drug for the treatment of rheumatoid arthritis that has shown potent anticancer activity against many panels of cancer cells, including breast cancer cells [13,14,15]. Currently, it is being evaluated in clinical trials for lung cancer (NCT01737502) and ovarian cancer (NCT03456700) [16]; moreover, its efficacy against the viral infection COVID-SARS-19 is also under study [17]. The molecular structure of Auranofin is shown in Figure 1 and it consists of a thiosugar ligand, the 2,3,4,6-tetra-O-acetyl-1-thio-β-D-glucopyranose, bound to the gold(I)triethylphosphane co-ligand.

Mechanism of action of Auranofin

As with other metal-based drugs, Auranofin exerts its biological activity thanks to ligand-exchange chemical properties, based on the binding affinity of the “soft” Lewis acid Au(I) atom, toward ‘‘soft’’ Lewis bases. Sulfur-containing amino acids represent the main chemical target for gold coordination, making Auranofin a potent inhibitor of enzymes whose active site bears cysteines or selenocysteine residues. Among them, thioredoxin reductase (TrxR) is endowed with a nucleophilic selenocysteine moiety in the active redox site, recognized as the main site for enzyme inactivation by gold-based drugs [16]. TrxR covalent inhibition is considered the most effective mechanism of Auranofin anticancer activity, with TrxR being one of the key enzymes controlling the redox homeostasis of cells. The imbalance of the protective redox systems by which cancer cells can keep under control their abnormally high reactive oxygen species (ROS) levels is a powerful strategy to activate apoptosis and promote cancer cell death [16]. A study on ovarian cancer cells has demonstrated that Auranofin exerts cytotoxic activity by increasing the production of ROS [18].

An important drawback of Auranofin’s mechanism of action, through covalent modification of biomolecules, is the lack of selectivity, since many proteins contain cysteine residues, which can be easily targeted, causing severe side effects in the treated patients.

While the inhibition of TrxR (both cytosolic TrxR1 and mitochondrial TrxE2) is recognized as the primary pharmaceutical effect, several off-target mechanisms of action of Auranofin have been reported, widening the range of pathways in which the antineoplastic activity of this drug can be involved. The cytotoxic effect of Auranofin by targeting the ubiquitin-proteasome system (UPS) has been investigated in breast cancer cells [19], leveraging the presence of a catalytic cysteine residue in the proteasome-associated deubiquitinase (Dub) active site. The results have confirmed that Auranofin directly inhibits the activity of Dub USP14 at supra-cytotoxic doses, causing broad accumulation of proteasome substrates. However, neither direct inhibition of 20S proteasome activity nor disassembly of proteasome subunits was ascribed to Auranofin.

A recent study [20] has reported a synergistic effect of Auranofin and vitamin C (VitC) in targeting TNBC cells by affecting both the glutathione and thioredoxin systems. The redox-based anticancer strategy of the Auranofin/VitC combination resulted to be linked to prostaglandin reductase 1 (PTGR1) expression levels, suggesting a role of PTGR1 as a potential predictive biomarker.

Li et al., have reported Auranofin’s anticancer activity through inhibition of the expression levels of key proteins in the PI3K/AKT/mammalian target of rapamycin (mTOR) pathway, in non-small lung cancer cells [21]. This mechanism of Auranofin action reduces cancer cells’ ability to adapt to the hypoxic microenvironment, by altering the expression levels of the hypoxia-inducible factor-1 (HIF-1) [22]. Inhibitors of the PI3K/AKT/mTOR pathway represent a promising strategy for anticancer therapy; some of these drugs, such as the mTOR inhibitors everolimus and temsirolimus, have been approved for the treatment of breast cancer [23]. As other gold(I) compounds, Auranofin (with IC_50_ ~ 4 μM) can inhibit key enzymes involved in glucose metabolism of cancer cells, in particular by glycolysis dysregulation through inhibition of hexokinase (HK) and phosphofructokinase (PFK), leading to ATP depletion [24]. These mechanisms of action by glycolysis inhibition have not been reported in breast cancer cells, but they might represent new fields of study.

#### 3.1.2. Phosphane Gold(I) Compounds

In vitro studies

Since the 1980s, when Auranofin was discovered to be a powerful anticancer compound, phosphane gold(I) compounds have been considered and tested over many panels of cancer cells [14,15]. The triethylphosphanegold(I) cation has been recognized as the active fragment; hence, Auranofin is essentially a prodrug [25]. However, the trialkylphosphanegold cations are extremely toxic, leaving many doubts about their involvement in the design of new gold drugs [26]. As concerns breast cancer cells, some efforts have been directed to the study of different phosphane moieties, mainly containing aromatic groups, to lower the toxicity despite enhanced or similar anticancer activities. Since the 1990s, biochemical studies have been led to determine the main mechanism of action of phosphane gold(I) compounds. The results, attained with mild differences in the chemical structures, were affected by extreme variability and they did not make a focus of interest in a restricted class of compounds possible. Some recent results in terms of IC_50_ are reported in Table 1 and, even though the values refer to different times of cell treatment, most of the phosphane gold(I) compounds resulted extremely active in regards to MCF7 (cisplatin-resistant cells) or MDA-MB-231 cells (in vitro models of luminal A and TNBC, respectively), similarly to cisplatin and Auranofin. The herein considered phosphane gold(I) compounds consist of linearly coordinated gold centers bound to phosphane ligands and to another co-ligand. The latter might be an additional phosphane ligand or one of a wide range of neutral or anionic ligands bearing C, N, S and O donor atoms, giving rise to cationic or neutral derivatives. Moreover, most of the works cited in Table 1 highlighted the strongest activity of the gold-based derivatives in comparison with the free phosphane ligands and co-ligands by themselves [27,28], regardless of the nature of the donor atom bound to the gold center.

Most of the compounds in Table 1 exhibited a very good inhibitory effect in regard to either mitochondrial or cytosolic TrxR, with the most active as anticancer drugs, as well as very potent as TrxR inhibitors [27,29,30,31]. Remarkably, the result for a trialkyl phosphane ligand displayed high activity in regards to HeLa cells (more than cisplatin) but much less so in respect to MCF7 cells [32], while trialkyl PCy_3_-Au-S derivatives (S = mercaptobenzoate ligands) appeared more active than Et_3_P-Au-S and Ph_3_P-Au-S analogs; moreover, although they strongly inhibited TrxR, it was found that they also activated the apoptosis process by p53 signaling in MCF7 cells [31]. The cellular uptake for phosphane gold(I) compounds was quite good, depending on the hydro/lipophilicity balance of the molecule; interestingly, for some P-Au-C compounds (C = alkynyl), the formation of fibrillar structural aggregate strongly hampered the cellular drug uptake and then the activity on MDA-MB-231 cancer cells [33]. Furthermore, the activity of P-Au-S derivatives was compared to that of NP-S-Au, where NP is for nanoparticle, regardless of NP size, to highlight the role of the phosphane ligand in the activity [34]. A really interesting upgrade on the search of new active gold phosphane compounds is the selective activity of compounds between healthy cells and sick breast cancer cells; a high selectivity factor (SF = normal cells IC_50_ divided by malignant cells IC_50_) decreases the systemic toxicity, without compromising their anticancer power, making the gold compounds candidates in the ambitious design to contrast breast cancer. In recent years, highly selective compounds with P-Au-Cl [30] or P-Au-S [35] chemical environments have been found to display very high selectivity factors (SF = 10–35 for the latter). Two chiral (e)-diphosphine-digold(I) complexes, containing mono- and di-methyl ester-substituted diphosphane ligands, are selective in regards to normal human breast epithelial cells (MCF10A cells) with a selectivity that is a function of (a) the concentration, (b) the presence of one or two methyl ester functionalities and (c) the chirality. The optical isomerism seems to also play a role in the anticancer activity toward breast cancer cells; the activities of RR or SS chiral phosphane gold(I) chloride compounds have been compared, revealing a substantial difference between the two enantiomers, likely due to a different lipophilicity, then affecting the cellular uptake [30].

One of the most interesting recent developments of research on anticancer phosphane gold compounds concerns a class of gold complexes containing acridines functionalized with thiourea, for example the 1-acridine-9-yl-methylthiourea, in addition to a PPh_3_ moiety [35]. Even though TrxR is recognized as the most likely molecular target for gold(I) compounds even in breast cancer cells, recent works have also taken into consideration DNA binding, whose intercalation represents the most likely mode of action adopted by cisplatin and related compounds. The gold complexes–DNA binding constants measured by UV-visible absorption or fluorescence competition assays using calf thymus DNA as a model, range between 1.55 and 6.12 × 10^3^ M^−1^ for azolate/phosphane gold(I) compounds [36], between 2.12 and 7.48 × 10^6^ M^−1^ for aminophosphane gold(I) compounds conjugated with different thiolates [29], and between 1.1 and 3.7 × 10^4^ M^−1^ for triphenylphosphane gold(I) conjugated with 1-acridin-9-yl-methylthiourea [35]. Leaving aside the second entry of compounds [29], whose affinity is comparable to that of classic DNA-intercalators, the other gold(I) compounds present affinity constants of one or two orders of magnitude lower than the binding constant recorded for ethidium bromide, K_EthBr_ = 1.2 × 10^6^ M^−1^, an inter-strand intercalating compound for DNA that strongly emits when excited at 520 nm. However, the high value of an intrinsic binding constant does not indicate a modification in the structure of gold(I) complexes, as highlighted by agarose electrophoresis and EMSA assays using the plasmid pEMBL9 and chromosomal DNA from *Anabaena* sp. PCC 7120, respectively; such evidence leads to the conclusion that, at least for the aminophosphane–Au–thiolate compounds, interaction with DNA does not introduce a large structural alteration in gold(I) compounds [29]. In acridine–AuPPh_3_ compounds, confocal microscopy and transmission electron microscopy locate the compounds in breast cancer cell nuclei; further, DNA viscosity measurements, as well as confocal microscopy observation, determine DNA binding through intercalation, exhibiting a dose-dependent response on topoisomerase I-mediated DNA unwinding. In addition, acridine–AuPPh_3_ compounds exhibit potent antiangiogenic effects and are also able to inhibit vasculogenic mimicry of highly invasive MDA-MB-231 cells [35].

The IC_50_ values reported in Table 1 were obtained for gold(I) phosphane compounds having different chemical coordination linking around the gold center and various ligands. In Figure 2, some representative molecular structures of the compounds quoted in Table 1 are reported.

In vivo studies

The most recent in vivo studies are substantially few if compared to the in vitro IC_50_ determinations. In fact, after the discovery, in 2011, that bis-phosphane chelated gold(I) compounds (i.e., [Au(dppe)_2_]Cl, where dppe is [1,2-bis(diphenylphosphane)ethane)-gold(I)]chloride) displayed significant in vivo antitumor activity against a range of murine tumor models, including both leukemia and solid tumors, but also severe side effects as liver, kidneys and systemic toxicities, research focused on diminishing both lipophilicity and chemical stability, considered liable for accumulation and mitochondrial dysfunction [44]. The design of gold(I) chelate compounds displaying more negative logP was attained by substituting phenyl groups in the phosphane ligand with pyridyl moieties and, when the bridged phosphine ligand was 1,3-bis(di-2-pyridylphosphane)propane (d2pypp), relative [Au(d2pypp)_2_]Cl was found to possess a good cytotoxic selectivity in regards to MDA-MB-468 breast cancer cells versus normal breast cells [45]. Unfortunately, most of the bis-phosphane gold(I) compounds, either chelated or linearly coordinated to carbenes or thiol ligands as in the example, have not been studied in vivo on models bearing breast cancers cells [45,46]. Recently, in 2019, a combination of vitamin C with Auranofin was applied on an in vivo study regarding MDA-MB-231 xenografts in mice; the study highlighted the absence of side effects and positive effects on the tumor volume reduction. Moreover, the available mixed drugs are active against several cancer cell lines and a comparative study identified, as a possible biomarker, PTGR1 (prostaglandin reductase 1) for the response of this treatment in TNBC [20]. The only recent work reporting in vivo studies of phosphane gold(I) compounds was conducted by Gambini et al. in 2018 [38]. After preliminary in vitro MTT tests on MDA-MB-231 and A17 breast cancer cells with various gold phosphane compounds having P-Au-N or P-Au-Cl backbones in the molecular structure, the most active compounds were used against A17 tumors transplanted in syngeneic mice. The A17 transplanted tumors share a molecular signature with basal-like breast cancer, including the expression of vimentin, cytokeratin 14, N-cadherin and Cox-2 [47]. Interestingly, the compounds having protic polar groups, such as COOH or OH, in the phosphane or in the azolate ligands, displayed a loss of cytotoxic activity in vitro. The antineoplastic activity of the selected compounds, 4,5-dichloro-imidazolate-1yl-gold(I)-triphenylphosphane (1) and 4,5-dicyano-imidazolate-1y-gold(I)-triphenylphosphane (2), and that of cisplatin were compared by analyzing data obtained on treated transplanted mice. The study started 10 days after tumor challenge by administering 3 mg/kg/day of each compound once every 3 days, 4 times, in accordance with the protocol q3X4. The gold compounds were found to inhibit the tumor mass increases, even though to a lower extent than cisplatin; however, unlike the latter, they seemed to not affect the overall wellness of the mice, avoiding fur and weight losses. The histopathological analysis and the quantitative analysis of metals in the explanted kidneys displayed a minor accumulation of gold with respect to platinum (Figure 1).

#### 3.1.3. NHC–Carbene Gold(I) Compounds

General

N-Heterocyclic Carbene ligands (NHC) are optimal ligands for gold(I) affording a class of gold(I) compounds that exhibit remarkable biological properties and their activity has been the focus of many recent reviews [48,49,50]. The interest in the biological application of these gold(I) compounds has steeply increased in the last few years.

Being excellent sigma donors with a planar coordinative environment bulked by the substituents on 1, 3 positions, the chemical and stereoelectronic features of NHC ligands resemble those of triaryl or trialkyl phosphanes [51,52].

In general, NHC gold(I) compounds might be divided into two classes, namely, the neutral and the cationic classes, depending on the nature of ligands bound to the gold(I) to accomplish the requirement of the linear coordination of Au(I); in fact, when X is an anion or a negatively charged ligand, the lipophilic structures A, B, C and D are obtained, while, when L is a neutral ligand, such as the carbene ligand by itself or, for example, a phosphane, the more hydrophilic cationic homoleptic E or the heteroleptic F compounds are obtained, respectively. In Figure 3, some selected structures for NHC derivatives are reported.

The design of NHC gold(I) compounds, according to the neutral or cationic classification, is the key point for this class of compounds as it concerns the hydro/lipophilic balance, hence the logP values, which govern, in some way, the mechanism of action and the drug’s selectivity toward healthy and sick cells [53,54]. Moreover, the nature of the co-ligand L or of the counterion X may vary hugely both in the chemical structure and the polarity, affording a wide range of anticancer NHC gold compounds, whose anticancer activities have been reviewed in 2018 da Porchia et al. [50]. Concerning the anticancer activity of NHC gold(I) derivatives against breast cancer, many in vitro studies have been applied on breast cancer cell panels and some of the most representative IC_50_ values are reported in Table 2. In general, the biological activity is featured by quite good cellular uptake [55], rather good stability in the physiological medium and strong anticancer activity, with recent examples which stand out concerning other gold(I) compounds or classic metal-based drugs currently in use, such as cisplatin.

In vitro studies

As reported in Table 2, most of the in vitro studies concerning NHC gold(I) compounds have been conducted on MCF-7 and MDA-MB-231 breast cancer cells. MCF-7 cells are ER-positive and PR-positive and belong to the luminal A molecular subtype. MCF-7 cells appear more differentiated and, normally, are considered poorly aggressive and non-invasive cells, whereas MDA-MB-231 cells belong to the triple-negative molecular subtype; they appear less differentiated with a mesenchymal-like morphology and a high metastatic potential. In some case, NHC gold(I) compounds have also been tested in other TNBC cell lines, such as MDA-MB-468, HS 578T and BT-549 cells.

In vivo studies

Even though many in vitro studies have been led to test the cytotoxic activities of gold(I) carbenes against numerous panels of cancer cells, only a few recent studies report in vivo tests and none of them is about breast cancer models. After an extensive in vitro study highlighting a mechanism of action mainly involving mammalian TrxR inhibition in the lower micromolar range, a preclinical report on a 2,3,4,6-tetra-O-acetyl-α-D-glucopyranosyl-1-thiolate derivative and 1,3-dibenzyl-4,5-diphenyl-imidazol-2-ylidene gold(I) dimethylamino dithiocarbamate describes prostate cancer growth inhibition on an advanced prostate cancer (PC3) xenograft model [58,64]. The anticancer effect of these compounds was associated with a good tolerability by mice in terms of weight loss and general wellness. An additional example of dinuclear gold(I) complex with Bisbis(N-heterocyclic Heterocyclic-Ccarbene) and diphosphine ligands tested in vivo is quoted for two independent animal models, i.e., HeLa xenografts (showing 81% inhibition of tumor growth after i.p. administration with 15 mgkg^−1^ once every 2–3 days for 8 days) and highly aggressive mouse B16-F10 melanoma, with no detectable side effects [46,65]. An in vivo study has reported a combined therapy with vascular-disrupting agents and gold(I) compounds of the type bis(NHC)gold(I) compounds displaying antimetastatic properties; high cytotoxicity in resistant cancer cells and excellent tolerance by animals were found. The lipophilic gold(I) bis-carbene complexes, consisting of imidazole with 1-(3,4,5-trimethoxy)-phenyl and 1-(4-methoxy)-phenyl as 4,5 substituents, affected F-actin and matrix metalloproteinases, stopped cells in the G1 phase and disrupted blood vessels [56]. A very strong antitumor activity was found in MCF7 cells, with sub-micromolar IC_50_ values (see Table 2); these compounds were also effective in vivo against a highly metastatic B16-F10 mouse melanoma xenograft model. The overall preliminary results are indicative of considerable in vivo antitumor activity associated with excellent tolerance of this class of compounds.

### 3.2. New Enzymatic Targets of Gold(I) Compounds: Dihydrofolate Reductase

Despite the chemical preference of the Au center for the thiols and seleno-thiol groups, a certain diversity in the spectrum of action of gold-based compounds has been reported, which can be associated with the diverse chemical structure of the bound moieties in gold complexes. X-ray crystal structures of adducts formed by target proteins and gold(I)-based drugs have revealed unexpected sites of interactions of Auranofin, gold phosphane compounds, or gold carbene complexes [66,67,68]. These studies have evidenced not only that gold binds to N atoms of lysine or histidine residues, besides the S atom of cysteines, but also that secondary binding can occur between the ligand moieties of these complexes and protein binding pockets [67]. There is evidence that the mode of action of gold-based metallodrugs can be more complex than expected. In a study performed in breast cancer cells, we have observed heterogeneity of actions of phosphane gold(I) compounds with respect to target proteins in exerting anticancer activity [36]. The compounds under study, 3,5-dichloroimidazolyl-gold(I)-triphenylphosphane and 3,5-dicyanoimidazolyl-gold(I)-triphenylphosphane, resulted to be very active against TNBC and HER2-positive breast cancer cells, by inhibiting both TrxR and dihydrofolate reductase (DHFR) enzymatic activity, in vitro. Based on these results, we can hypothesize this multitargeted mechanism to be responsible for the previously observed antineoplastic effect in vivo [38]. DHFR does not have selenocysteine residues and the available thiol groups belonging to cysteines are not involved in the catalytic site. The product of DHFR catalyzed reaction, tetrahydrofolate (THF), is essential for de novo synthesis of purine and thymidylate in cell proliferation. Inhibition of DHFR activity results in depletion of fundamental metabolites for cell proliferation, making this enzyme an attractive target in drug design for the development of chemotherapeutics molecules. Currently, the most powerful DHFR inhibitor acting as an anticancer drug is the folate analogous methotrexate (MTX), whose efficacy is affected by relevant dose-related side effects and the development of resistance, as observed in the MCF7 breast cancer cell line [69,70]. For this reason, great effort is addressed to the discovery of more selective and effective bioactive compounds acting as inhibitors of this enzyme [71]. Drug design targeted to DHFR to block cancer cells proliferation has developed in the last years with the aim to enhance the cellular uptake and reduce side effects. One of the newly developed anti-cancer agents is pralatrexate, a 10-deazaaminopterin analogous of methotrexate, approved by FDA and EMA in 2009 [72]. In terms of anti-tumor activity, pralatraxed has shown IC_50_ values very close to or even lower than methotrexate, as reported in high-risk neuroblastoma cells, in which the IC_50_ of pralatrexate was approximately ten-fold less than the IC_50_ of methotrexate [73]. Non-classical antifolates are lipophilic molecules, which do not need folate transport systems and are easily internalized by passive diffusion into cells. Several classes of this kind of DHFR inhibitors recently synthesized have been found to possess good antineoplastic activity, most of them being characterized by the presence of heterocyclic moieties in the structure, such as the novel 6-substituted pyrido[3,2-d]pyrimidines with a three- to five-carbon bridge. Among these compounds, the unsubstituted saturated three-carbon-bridged analogous (6-(3-phenylpropyl)pyrido[3,2-d]pyrimidine-2,4-diamine) showed the highest cytotoxicity against HL-60 cells with an IC_50_ = 0.20 μM and resulted to strongly inhibit recombinant human DHFR, with an IC_50_ value of 0.06 μM [74]. Some 6-substituted pyrrolo[3,2-d]pyrimidines acting as dual inhibitors on thymidylate synthase (TS) and DHFR had been previously designed and tested in cancer cells [75]. Notably, future anti-tumor strategies appear to rely on the multi-target approach of antifolates targeting efficiently DHFR and folate receptors, as well as other enzymes, such as thymidylate synthase or TrxR. Thanks to their structural features, some gold(I) phospane drugs endowed with an inhibitory affinity for DHFR [76], but also for other proteins, besides TrxR, seem to comply with this new generation of anticancer drugs [36].

## 4. Conclusions

Breast cancer is the most common cancer in women worldwide. Early detection permits survival probabilities higher than 90% for at least 5 years after diagnosis to be achieved, but this percentage drops to 27% in case of metastatic disease. Thus, the development of new therapeutic strategies is still of crucial importance. In this review, we take into account the last works on two main categories of gold compounds, those containing phosphane groups and those containing carbene groups. Both of them exhibit very strong in vitro activity against breast cancer cell lines; furthermore, some studies have highlighted their sparing ability towards healthy cells. Additionally, in most of the compounds herein considered, the ligands are not active and the introduction of gold activates their anticancer activity. Moreover, despite the heterogeneity of the molecular structures, displaying mono- or poly-nuclear organizations, ionic or neutral assemblies and a variety of different ligands, both classes share the recognition of similar main targets. Although TrxR resulted to be the most strongly recognized and the most widely studied target, studies on residual activity on cancer cells highlighted other protein targets, not only characterized by the presence of cysteine residues in the catalytic site. From the surveyed studies, it emerged a strong and diverse impact of the ligands bound to the Au center in tuning the binding affinity of the gold complexes towards molecular targets, resulting in a more complex mechanism of action than selective auration of the thiols or seleno-thiol protein groups. The emerged multi-target action of gold-based complexes may be useful for cancer therapy. Simultaneous inhibition of TrxR, which results in accumulation of intracellular ROS and of other target enzymes which play crucial role in cell proliferation and in cell metabolism, such as DHFR or glycolytic enzymes (PKC, HK, etc.), can exert a multilateral effect in inducing apoptosis of breast cancer cells. Multi-target therapeutic strategies involving a gold-based drug have been recently reported in the literature, e.g., the combined treatment with Auranofin and trametinib, a mitogen-activated protein kinase (MAPK) inhibitor, has exhibited a synergistic effect inducing cell death in MCF-7 human breast cancer cells [77]. In this scenario, we can hypothesize that novel gold(I) compounds with multiple targets might represent a valuable strategy in drug design against breast cancer. The development of new gold-based targeted drugs can also take advantage of novel bioactive and luminescent gold(I) complexes able to outline the biodistribution of gold-containing compounds by bioimaging approaches [78]. In particular, luminescent molecules belonging to the phosphane gold(I) class have been reported to display a good IC_50_ in tumor cells, as well as a good cellular internalization, with uniform biodistribution in the cytoplasm of MDA-MD231 and MCF-7 cells when analyzed by confocal microscopy [79]. Thus, gold (I) complexes can be also considered as an attractive molecular tool to track bioactive molecules inside tumor cells.

## Data Availability

Not applicable.

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
