# Peer review of "Breast Cancer Treatment: The Case of Gold(I)-Based Compounds as a Promising Class of Bioactive Molecules"

_biomolecules, 2022, doi:10.3390/biom12010080_

Round 1

Reviewer 1 Report

Galassi, Marchini and collaborators present in the manuscript a revision on the Gold(I) compounds as promising new metallodrugs with bioactivity specially for breast cancer cell lines. The manuscript is well organized, and has potential interest for the bioinorganic community around the world. The manuscript should be accepted for publication, my only reservation is described below:

  1. The main articles, which bring information about this research topic, were gathered in this review, although some recently published articles must be cited to make it more complete concerning the application of confocal microscopy as an example of tool to track bioactive molecules inside of tumor cell. Bioactive and luminescent molecules belonging to the phosphane Gold(I) class, were prepared and applied as markers by Pope et al ( Chem. Commun.  2014 (50) 10343) and Favarin et al ( New Journal of Chemistry 2020 (44) 6862-6871). The molecules presented by these authors presented bioactivity and were capable to be tracked by confocal microscopy inside human breast adenocarcinoma cell lines and other and also presented good IC50.      
  2. Iten 3.1.2 it is Phosphane instead of Phophane

Thank you for the contribution to the Gold(I) chemistry.

Author Response

Galassi, Marchini and collaborators present in the manuscript a revision on the Gold(I) compounds as promising new metallodrugs with bioactivity specially for breast cancer cell lines. The manuscript is well organized, and has potential interest for the bioinorganic community around the world. The manuscript should be accepted for publication, my only reservation is described below:

  1. The main articles, which bring information about this research topic, were gathered in this review, although some recently published articles must be cited to make it more complete concerning the application of confocal microscopy as an example of tool to track bioactive molecules inside of tumor cell. Bioactive and luminescent molecules belonging to the phosphane Gold(I) class, were prepared and applied as markers by Pope et al ( Chem. Commun.  2014 (50) 10343) and Favarin et al ( New Journal of Chemistry 2020 (44) 6862-6871). The molecules presented by these authors presented bioactivity and were capable to be tracked by confocal microscopy inside human breast adenocarcinoma cell lines and other and also presented good IC50.      
  2. Iten 3.1.2 it is Phosphane instead of Phophane

Thank you for the contribution to the Gold(I) chemistry.

  1. We thank the Reviewer for the positive evaluation of our manuscript and for her/his valuable suggestions to improve it. We have now cited and commented on the papers by Pope et al. and Favarin et al. in the Conclusion section. In addition, we have corrected the “Phosphane” word as indicated.

Reviewer 2 Report

The review-manuscript entitled “Breast cancer treatment: the case of Gold(I)-based compounds as a promising class of bioactive molecules” by Galassi et al.

Authors in this review covered studies on the anticancer properties of gold(I) complexes against different types of breast cancer, highlighting the mechanism of actions and structural-property relationship. The review is well-structured and good addition to the scientific community. However, there are few issues need to be addressed before accepting the manuscript for publication:

[1] Although, I am not a native English speaker, there are occasions where I found the sentences needs some polishing such as lines 206-209. Moreover, some sentences were extremely long and hard to understand without revisiting the sentence twice. I recommend language checking by a native English speaker.

[2] The structures of the ruthenium complexes highlighted in page 3 need to be illustrated somewhere in the manuscript to aid the reader.

[3] There are four IC50 values listed for P-Au-P in Table 1. I checked the reference and found that there are 4 different compounds, but they are not illustrated in Scheme 2. I would recommend listing all the different structures or removing three of the IC50. The same applies for the rest of Table 1 and Table 2.

[4] Auranofin is listed twice IN table 1 from two different references. It is quite normal to encounter variations in IC50 from lab to lab. Therefore, cross-comparison anticancer data from different labs is not a good practice. I found throughout the manuscript that authors highlighted comparison between a particular compound and a reference drug. Hopefully, this comparison was made for data from the same reference.

[5] The references have double numbers in the list.

Regards,

Author Response

Reviewer 2

Comments and Suggestions for Authors

The review-manuscript entitled “Breast cancer treatment: the case of Gold(I)-based compounds as a promising class of bioactive molecules” by Galassi et al.

Authors in this review covered studies on the anticancer properties of gold(I) complexes against different types of breast cancer, highlighting the mechanism of actions and structural-property relationship. The review is well-structured and good addition to the scientific community. However, there are few issues need to be addressed before accepting the manuscript for publication:

[1] Although, I am not a native English speaker, there are occasions where I found the sentences needs some polishing such as lines 206-209. Moreover, some sentences were extremely long and hard to understand without revisiting the sentence twice. I recommend language checking by a native English speaker.

We thank the Reviewer for the recommendation about language checking. We have revised the manuscript carefully and some long sentences have been rewritten hopefully making them easier to read.

[2] The structures of the ruthenium complexes highlighted in page 3 need to be illustrated somewhere in the manuscript to aid the reader.

  1. Although the anticancer activities of some representative ruthenium compounds have been reported in the introduction, to highlight the interest toward metal-based drugs, actually the focus of the present review is the gold (I) complexes and their anticancer properties, thus we decided to not show ruthenium complexes structures, that anyway are available in the reported literature.

[3] There are four IC50 values listed for P-Au-P in Table 1. I checked the reference and found that there are 4 different compounds, but they are not illustrated in Scheme 2. I would recommend listing all the different structures or removing three of the IC50. The same applies for the rest of Table 1 and Table 2.

  1. Both Table 1 and Table 2 have been revised according to the Reviewer’s suggestions and we decided to remove the IC50 values of the compounds not reported in Scheme 2. Moreover, Scheme 2 has been modified and now it includes new compounds.

[4] Auranofin is listed twice in table 1 from two different references. It is quite normal to encounter variations in IC50 from lab to lab. Therefore, cross-comparison anticancer data from different labs is not a good practice. I found throughout the manuscript that authors highlighted comparison between a particular compound and a reference drug. Hopefully, this comparison was made for data from the same reference.

  1. We revised the Table 1 removing one of the two references related to Auranofin and the corresponding IC50 values, as suggested by the Reviewer, maintaining the most representative Auronofin IC50. We have also eliminated the Auranofin in table 2 where only carbene compounds have been reported. Of course, when a comparison between a particular compound and a reference drug has been done, data from the same reference were considered.

[5] The references have double numbers in the list.

  1. Thanks for the remark. We realized that probably this problem arises because this manuscript has been uploaded by using a Macintosh pc. We will try to solve this issue before publication.
